# Machine learning-based prediction of the axial load capacity of UHPC strengthened reinforced concrete columns: A comparative analysis

Viet Hai Hoang[1]*, Minh Quang Tran[2], Van Thuc Ngo[3]

1 Faculty of Civil Engineering, University of Transport and Communications, Hanoi, Vietnam, 2 University of Minho, Guimaraes, Portugal, 3 Mien Tay Construction University, Vinh Long, Vietnam

* hoangviethai@utc.edu.vn

## Abstract

This study develops and evaluates machine learning (ML) models to predict the axial load capacity (Pu) of reinforced concrete (RC) columns strengthened with ultra-high-performance concrete (UHPC) jackets. A comprehensive experimental database containing 105 test samples with 17 key input parameters was compiled from the literature, representing the most extensive dataset of UHPC-jacketed RC columns to date. Using this database, a machine learning (ML) framework was established to predict the ultimate axial load capacity, employing six models: Extremely Randomized Trees (ER) model, K-Nearest Neighbors (KNN), Light Gradient Boosting Machine (LightGBM), Xgboost, CatBoost, and Cascade Forward Neural Networks (CFNNs). The CatBoost model achieved the best performance with $R^2 = 0.983$, MAE = 177 kN, and RMSE = 211 kN, significantly outperforming traditional design codes such as ACI 318 and EC2. In addition to high predictive accuracy, SHAP analysis was conducted to interpret the influence of each parameter, providing new insights into the mechanical behavior and governing factors of UHPC-jacketed RC columns. These findings highlight the capability of advanced ML to capture complex nonlinear effects more effectively than traditional methods. The proposed framework not only provides new insights into the mechanics of UHPC–RC columns but also offers a reliable predictive tool to support safer and more efficient design for strengthening.

## 1. Introduction

Reinforced concrete (RC) columns are important load-bearing structures in buildings and bridges. After many years of service, many existing RC columns often encounter problems of reduced load-bearing capacity due to material strength loss, design defects, aging, or increased load demands. To extend their service life, rehabilitation techniques have been widely applied. Some common techniques include steel

**Data availability statement:** All relevant data are within the paper and its Supporting information files.

**Funding:** This research is funded by the University of Transport and Communications (UTC) [grant number T2025-CT-004TD to V.H.H.].

**Competing interests:** The authors have declared that no competing interests exist.

jacketing, fiber-reinforced polymer (FRP) confinement, and concrete encasement. Ultra-high-performance concrete (UHPC) has emerged as a transformative material in structural engineering. UHPC is distinguished by its outstanding compressive strength, tensile ductility, and long-term durability [1,2]. Not only is it used in new construction, but UHPC is also rapidly expanding and being used in reinforcing existing infrastructure. UHPC has proven to be superior to conventional materials in repair works. It increases toughness and improves load-bearing capacity significantly compared to traditional concrete materials. In recent years, UHPC has emerged as a promising retrofitting material [3–6]. When used as an external jacket, UHPC can provide substantial confinement and strength enhancement to existing RC columns [7]. However, despite these advantages, design codes such as ACI 318 and Eurocode 4 do not provide explicit provisions for UHPC-jacketed RC columns, leading to uncertainty in practical design and assessment.

Many studies have focused on studying the axial and combined load responses of UHPC reinforced RC columns, and certain achievements have been made [8,9]. Experimental studies have demonstrated that UHPC cladding can significantly improve both axial load capacity and deformation performance. This is achieved by enhancing the overall material strength and delaying the occurrence of premature failure [10]. However, these studies are scattered across different laboratories using a variety of specimen geometries and material properties, and often lack unified predictive models. Conventional analytical methods, adapted from existing concrete or composite design equations, tend to oversimplify the limiting mechanism and provide limited accuracy when applied to UHPC envelopes. In particular, the design frameworks often extend existing limit models for high-strength and conventional concrete. However, such provisions tend to rely on simplified assumptions about stress–strain relationships and uniform confinement. It fails to fully capture the nonlinear and localized behavior observed in UHPC–RC composite systems. This gap underscores the need for more generalizable predictive models that can accommodate both material heterogeneity and complex interaction mechanisms.

Conventional analytical and numerical methods are thus challenged in two respects: (1) insufficient robustness across diverse column geometries and loading conditions, and (2) the inability to fully capture nonlinear interactions between UHPC, reinforcement, and existing concrete. These limitations motivate the need for more reliable, data-driven approaches to predict axial performance and ensure safe, economical design of UHPC-strengthened columns.

Emerging as a versatile tool in the 4.0 technology era, machine learning (ML) has recently developed strongly and been widely applied in civil engineering [11,12]. ML provides effective tools for exploring complex relationships in large, heterogeneous datasets [13]. A sufficiently powerful ML model is capable of accurately predicting outputs with less dependence on simplifying assumptions [12,14,15]. In the problem of determining the load-bearing capacity of UHPC-reinforced concrete columns, ML models are particularly well-suited for identifying hidden patterns between parameters such as shell thickness, reinforcement ratio, fiber volume, and concrete strength – interactions that are difficult to quantify with conventional models. However, to date,

only a few studies have explored the use of ML to predict the ultimate load-bearing capacity of UHPC reinforced concrete columns.

Katlav [16] applied ten ML algorithms to predict the moment-carrying capacity of hybrid beams consisting of UHPC–NSC, demonstrating superior accuracy compared to traditional confinement equations. Similarly, to predict the flexural capacity of reinforced UHPC beams, Taffese [17] employed an explainable machine learning model based on CatBoost regression, outperforming traditional mechanical models and six benchmark ensemble methods. Feature analysis identified beam height, longitudinal reinforcement ratio, and beam width as the most influential parameters, with interactions showing that fiber content above 2% amplifies the effect of the reinforcement ratio. These studies demonstrate the potential of explainable ML for accurate and interpretable predictions, but challenges remain due to the "black box" nature of some models and the limited availability of comprehensive datasets [18,19].

The above studies highlight the potential of explainable machine learning models in structural engineering by providing deeper insights into the decision-making mechanisms of complex algorithms. Nevertheless, the limited size of available experimental databases restricts their applicability and poses a significant challenge for reliable predictions. Despite the remarkable mechanical advantages and increasing use of UHPC in strengthening applications, RC columns strengthened with UHPC jackets have received very limited attention in the ML-based prediction domain. The existing empirical and analytical models for UHPC-jacketed columns are mostly derived from simplified assumptions and relatively small experimental datasets, which restrict their ability to account for the nonlinear interactions between geometry, material strength, and confinement effects. Furthermore, most prior ML studies have focused on conventional or FRP-strengthened columns, leaving a gap in understanding the predictive behavior and key influencing factors of UHPC-jacketed RC columns. A careful review of the literature reveals several critical research gaps that motivate the present study:

(1) Lack of ML studies for UHPC-strengthened RC columns: While ML has been widely used for predicting the capacity of conventional or FRP/steel-jacketed RC members, no systematic study has yet addressed RC columns strengthened with UHPC jackets, despite their growing use in retrofitting and rehabilitation

(2) Limited and fragmented experimental data: Existing analytical and empirical models for UHPC-jacketed columns are based on small, scattered experimental datasets, making them inadequate to capture the combined influence of geometry, material properties, and confinement effects

(3) Lack of comparative assessment of advanced ML algorithms: Previous ML studies rarely provide a comprehensive comparison among different state-of-the-art algorithms for this type of structural system

(4) Insufficient interpretability of ML predictions: Most existing ML-based studies focus only on prediction accuracy, with limited effort to interpret the influence of individual input parameters on the output response, which limits the physical understanding and practical use of the models.

The objective of this study is to develop accurate predictive models for estimating the axial load-bearing capacity of reinforced concrete (RC) columns strengthened with UHPC. The workflow of the study is summarized as follows:

1. Establishment of the first dedicated experimental database for UHPC-jacketed RC columns. The database includes 17 input parameters and one output feature.

2. Several machine learning models, including Extremely Randomized Trees (ER), K-Nearest Neighbors (KNN), LightBGM, XGBoost, CatBoost, and Cascade Forward Neural Networks (CFNNs), were constructed. The hyperparameters of these models were optimized using a grid search combined with 10-fold cross-validation to ensure robustness.

3. The predictive performance of the ML models was further assessed using the coefficient of determination ($R^2$), Mean Absolute Error (MAE), Mean Absolute Percentage Error (MAPE), and root mean square error (RMSE).

4. SHAP analysis was conducted to interpret the optimal model by examining feature importance and sensitivity, thereby enhancing transparency and providing engineering insights into the influence of input parameters on axial strength prediction.

5. Critical comparison with existing code-based equations, highlighting their limitations and suggesting directions for incorporating UHPC retrofits into future standards.

## 2. Methodological background

### 2.1 Experimental database

This study aims to develop predictive models for estimating the axial load-carrying capacity of reinforced concrete (RC) columns strengthened with ultra-high-performance concrete (UHPC) jackets. Accurate prediction of the capacity of such retrofitted members is of great importance in structural engineering, as it underpins safer and more cost-efficient design solutions. To this end, an extensive literature survey was conducted to compile experimental evidence on UHPC-jacketed RC columns. The database used in this study was compiled from fourteen published experimental studies on reinforced concrete (RC) columns strengthened with ultra-high-performance concrete (UHPC) jackets and tested under axial compression. Each record corresponds to one tested specimen and includes seventeen variables describing geometric, material, and strengthening parameters. To ensure data reliability and consistency, a systematic preprocessing procedure was adopted. All variables were converted into consistent SI units (mm, MPa, and kN), and parameter definitions were standardized across different sources. Data with missing essential information (such as UHPC strength, reinforcement ratio, or load capacity), ambiguous test conditions, or combined loading effects were excluded. Minor secondary parameters were estimated only when justified and clearly documented in the original references. Each data entry was cross-verified with the corresponding tables and figures from the source publication to ensure accuracy, and duplicates from overlapping datasets were removed. After applying these criteria and quality control procedures, a total of 105 complete and reliable specimens were retained, representing the most comprehensive and consistent experimental dataset available for developing and validating the proposed machine learning models. Based on this review, a novel and comprehensive database was systematically established by consolidating information from fourteen published sources [7,9,20–31]. This dataset provides, for the first time, a unified foundation for investigating the behavior of UHPC-retrofitted RC columns. The compiled data were subsequently processed and analyzed using machine learning techniques to ensure reliable capacity prediction. A structured feature engineering procedure was employed to refine raw variables and transform them into suitable input parameters for model training. For predictive modeling, six advanced algorithms widely adopted in structural engineering were utilized, namely ER, KNN, LightGBM, XGBoost, CatBoost, and CFNN. The subsequent sections present details on dataset preparation, feature selection, and the adopted machine learning models.

The collected specimens exhibit variability in several critical parameters, including column dimensions, longitudinal and transverse reinforcement ratios, jacket thickness, compressive strength of both UHPC and the original concrete, yield strength of reinforcing steel, as well as the fiber volume fraction and aspect ratio within the UHPC matrix. A schematic illustration of a typical UHPC-strengthened RC column and its defining parameters is shown in Fig 1, while a detailed summary of the experimental dataset is presented in Table 1.

In this study, a comprehensive dataset was established based on experimental investigations of RC columns retrofitted with UHPC jackets. Seventeen representative input features were considered as predictors of the ultimate axial load capacity ($P_u$) of UHPC-strengthened RC columns, comprising one categorical and sixteen numerical variables. The categorical feature was the cross-sectional type (CS), while the numerical features included: column width (b), column length (a) for rectangular sections or diameter (D) for circular sections, column height (h), sectional area of normal concrete ($S_{NC}$), compressive strength of normal concrete ($f'_c$), longitudinal reinforcement ratio ($p_t$) and transverse reinforcement ratio ($p_v$) in normal concrete, sectional area of UHPC ($S_{UHPC}$), longitudinal reinforcement ratio ($p_{t\ UHPC}$) and transverse

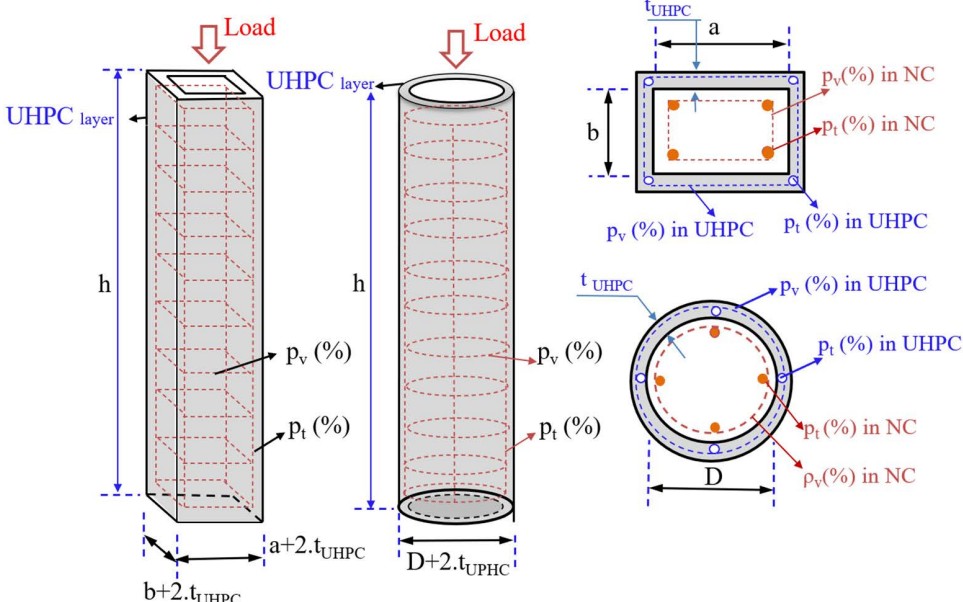

**Fig 1. Schematic and related structure of NC rectangular and circular columns strengthened with UHPC.**

**Table 1. Statistical information for parameters in the databases.**

| No | Feature | Type | Units | Min | Max | Mean | Median | Std |
|----|---------|------|-------|-----|-----|------|--------|-----|
| 1 | a | X1 | mm | 100 | 300 | 152.17 | 150 | 49.03 |
| 2 | b | X2 | mm | 100 | 300 | 158.50 | 150 | 49.31 |
| 3 | D | X3 | mm | 82 | 300 | 162.86 | 142 | 76.16 |
| 4 | h | X4 | mm | 300 | 1000 | 527.90 | 500 | 199.74 |
| 5 | $S_{NC}$ | X5 | mm² | 0 | 90000 | 25643.60 | 18145.84 | 20197.77 |
| 6 | $fc'_{NC}$ | X6 | MPa | 22.2 | 49 | 32.86 | 28.00 | 9.07 |
| 7 | $p_t$ in NC | X7 | % | 0 | 3.14 | 1.45 | 1.42 | 1.07 |
| 8 | $p_v$ in NC | X8 | % | 0 | 3.66 | 0.92 | 0.54 | 0.89 |
| 9 | $S_{UHPC}$ | X9 | mm² | 0 | 54977.87 | 11500.73 | 9974.56 | 11379.31 |
| 10 | $p_t$ in UHPC | X10 | % | 0 | 3.81 | 0.64 | 0.00 | 1.05 |
| 11 | $p_v$ in UHPC | X11 | % | 0 | 3.66 | 0.35 | 0.00 | 0.68 |
| 12 | $f_{yt}$ | X12 | MPa | 255 | 644 | 487.79 | 502.00 | 98.32 |
| 13 | $f_{yv}$ | X13 | MPa | 240 | 1173 | 524.33 | 454.00 | 292.62 |
| 14 | $fc'_{UHPC}$ | X14 | MPa | 81.6 | 189.97 | 115.88 | 116.08 | 19.73 |
| 15 | % fiber | X15 | % | 0 | 2.3 | 0.98 | 1.00 | 0.95 |
| 16 | $t_{UHPC}$ | X16 | mm | 0 | 50 | 18.50 | 20.00 | 13.59 |
| 17 | $P_u$ | Y | kN | 294.2 | 8100 | 1894.13 | 1381.99 | 1537.15 |

Note: Min. = Minimum; Max. = Maximum; Mean = Average; Median; Std. = Standard deviation.

reinforcement ratio ($p_{v\,UHPC}$) in the UHPC jacket, yield strength of longitudinal ($f_{yt}$) and transverse ($f_{yv}$) reinforcement, UHPC compressive strength ($f'_{c\,UHPC}$), UHPC jacket thickness ($t_{UHPC}$), and fiber dosage (volume fraction of steel fibers, % fiber) in the UHPC matrix as shown in Figs 3 and 4.

The dataset includes one categorical feature representing different concrete cross-sectional shapes. To enable its use across various machine learning models, such as ER, LightGBM, XGBoost, and CFNN, one-hot encoding was applied to convert this non-numeric feature into a numerical format. This transformation not only ensures compatibility with a wide range of algorithms but also enhances the interpretability of feature importance. Each category is represented by a separate binary column, where 1 indicates the presence of the category and 0 its absence. Fig 2 illustrates the CS feature before and after applying one-hot encoding.

The target variable in this study is the ultimate load-carrying capacity ($P_u$), defined as the maximum resistance attained by the strengthened column during experimental testing. Table 1 and Fig 2 present the statistical profiles of the input and output parameters of numerical features, reflecting both their variability and representativeness. The compressive strength of normal concrete ($f'_c$) generally falls within 22.2–49 MPa, while UHPC compressive strength ($f'_{c,UHPC}$) varying from 81.6 MPa to 189.97 MPa. Reinforcing steel exhibits yield strengths between 240 and 1173 MPa, the UHPC jacket thickness ranges from 0 to 50 mm, and fiber volume fractions vary from 0% to 2.3%. This heterogeneity in material and geometric characteristics establishes a solid basis for developing machine learning–based models to predict the axial performance of UHPC-strengthened RC columns.

## 2.2 Methodology

In this study, several ML algorithms were employed to develop predictive models for estimating the axial load-carrying capacity of RC columns strengthened with UHPC jackets. The selected algorithms represent diverse learning mechanisms, including ensemble-based tree models, instance-based learning, gradient boosting, and neural networks. This diversity enables a comprehensive evaluation of the nonlinear interactions between geometric, material, and reinforcement parameters that affect the axial load-bearing capacity of UHPC-encased RC columns. Extremely Randomized Trees (ER), K-Nearest Neighbors (KNN), Light Gradient Boosting Machine (LightGBM), Extreme Gradient Boosting (XGBoost), CatBoost, and Cascade Forward Neural Network (CFNN) are the six machine learning models that were employed in this study. For nonlinear regression tasks, each algorithm has unique benefits:

(1) ER is an ensemble technique based on bagging that reduces variance and overfitting by creating multiple randomized decision trees.

(2) KNN is an instance-based algorithm that captures local relationships between similar samples.

(3) LightGBM is a gradient boosting model designed for structured data, offering high efficiency and scalability.

(4) XGBoost is a powerful boosting algorithm with strong regularization capabilities that can effectively manage intricate nonlinear dependencies.

(5) CatBoost is an enhanced gradient boosting framework that reduces overfitting and effectively handles categorical variables.

(6) CFNN is a feedforward neural network extension that enhances learning stability and convergence by introducing direct connections from the input to deeper layers.

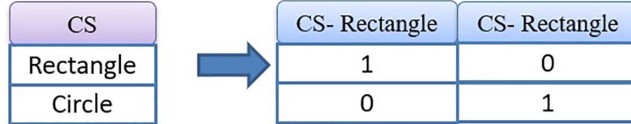

**Fig 2. One-hot encoding applied to the categorical Cross section (CS) feature.**

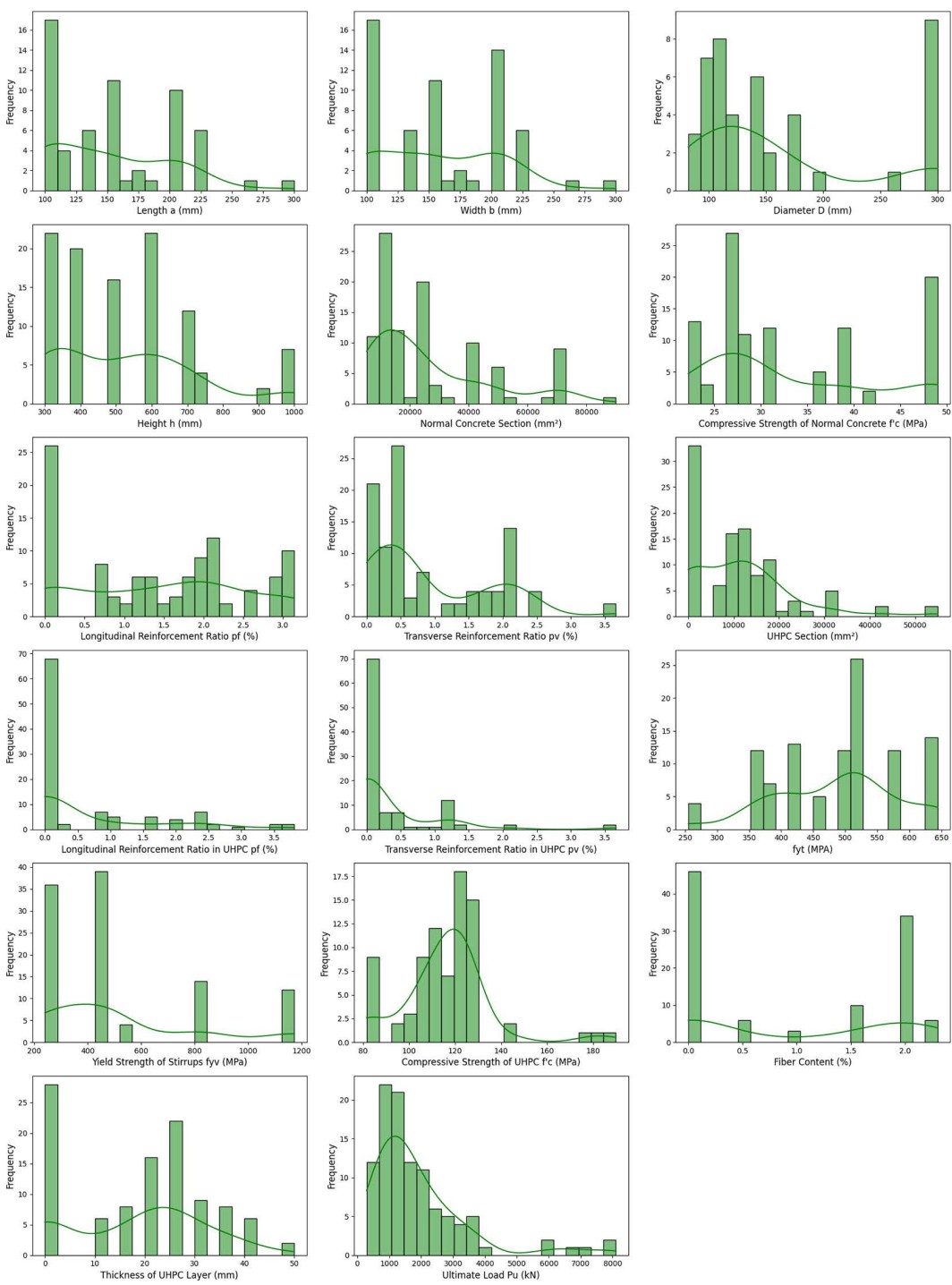

**Fig 3. Distribution of 16 numerical input and 01 numerical output features.**

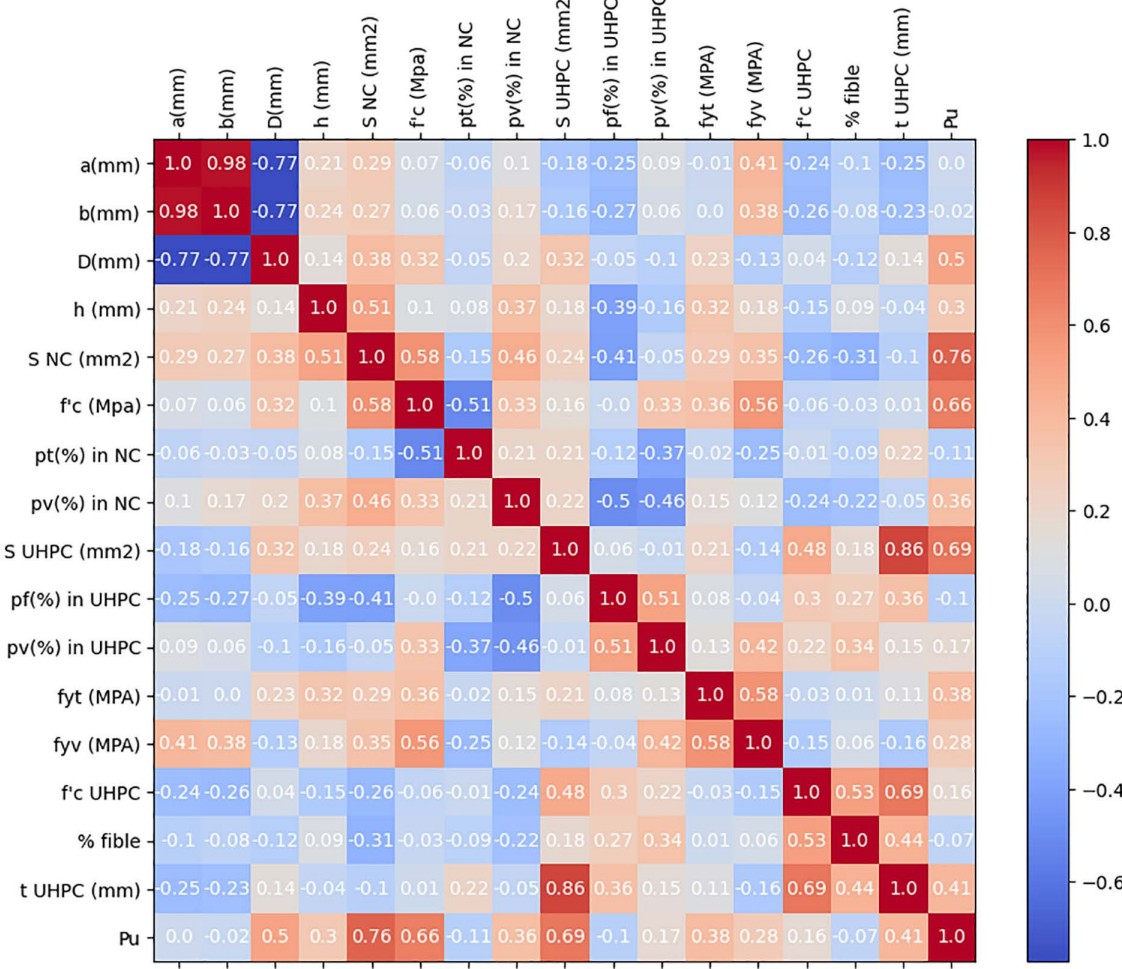

**Fig 4. Correlation matrix for the numerical input parameters of the dataset.**

These models were selected to ensure a broad representation of various learning mechanisms, including gradient boosting, bagging, kernel-based regression, and instance-based learning. Preliminary analyses showed that these algorithms provided a good balance between prediction accuracy, generalization, and computational efficiency for the given dataset. Other models, such as linear regression and artificial neural networks, were not included because their performance was found to be less stable or less interpretable for the relatively small dataset used in this study. The following subsections provide a concise overview of the theoretical background and implementation of each algorithm.

**2.2.1 Extremely Randomized Trees (ER).** The Extremely Randomized Trees (ER) algorithm is an ensemble technique that constructs a large number of decision trees and aggregates their outputs for prediction. However, where feature splits are chosen based on optimal criteria, ER introduces higher randomness by selecting both the features and the cut points at random during tree construction [32]. This strategy accelerates training and increases model diversity, which can enhance generalization performance. For regression problems, predictions are obtained by averaging the outputs of all trees, while classification relies on majority voting. Owing to its simplicity and resistance to overfitting, ER has been widely applied in various predictive modeling tasks.

**2.2.2 K-Nearest Neighbors (KNN).** The K-Nearest Neighbors (KNN) algorithm is a fundamental machine learning method commonly applied in both classification and regression tasks [33,34]. KNN works on approximation: the similarity between the query instance and existing data points is measured, often using the Euclidean distance. The algorithm then identifies the k closest neighbors from the training set and infers the output by aggregating their values – through majority voting in classification or averaging in regression. Its simplicity and intuitive design make KNN a widely recognized baseline model in predictive analytics.

**2.2.3 Extreme Gradient Boosting (XGBoost).** Extreme Gradient Boosting (XGBoost) is an optimized implementation of the gradient boosting framework. XGBoost is specifically designed for speed and performance. It builds predictive models by sequentially adding weak learners, typically decision trees. Each new learner focuses on correcting the residuals of the previous ones. XGBoost integrates advanced regularization techniques, such as L1 and L2 penalties, help control overfitting [35]. It has been extensively adopted for regression and classification tasks in both academia and industry.

XGBoost is suitable for the problem of predicting the axial resistance of UHPC reinforced RC columns because the relationship between input factors (concrete strength, core area, UHPC jacket area, reinforcement ratio, jacket thickness) and ultimate load Pu is strongly nonlinear and has complex interactions. Traditional methods often have difficulty capturing multiple relationships simultaneously. XGBoost is able to learn high-order interactions from data, control complexity through regularization mechanism, and make good use of second-order gradient information to make more stable and accurate predictions. The model not only predicts Pu effectively but also maintains generality when applied to diverse column configurations.

**2.2.4 CatBoost.** CatBoost [36] is a gradient boosting algorithm that was developed with a particular emphasis on handling categorical variables efficiently. CatBoost applies an innovative technique to encode categorical features and prevents prediction shift during training. It also incorporates ordered boosting and symmetric tree structures, leading to enhanced generalization capability and reduced overfitting. CatBoost has shown competitive performance across various regression and classification problems, especially when datasets contain a large proportion of categorical attributes.

In predicting the axial resistance of UHPC reinforced RC columns, CatBoost is useful because the data may contain many complexes, non-linear interacting features (between core area, UHPC jacket area, material strength, and reinforcement ratio). CatBoost can capture these nonlinear relationships well, while limiting prediction errors and maintaining high generalization ability. This explains why CatBoost often outperforms other boosting models in many applied studies.

**2.2.5 Light Gradient Boosting Machine (LightGBM).** LightGBM (Light Gradient Boosting Machine) [37] is an efficient and widely used supervised learning algorithm belonging to the ensemble family. LightGBM builds a series of decision trees sequentially, where each new tree is trained to minimize the residual errors of the existing ensemble. The contribution of each tree is scaled by a learning rate to prevent overfitting. This additive process, guided by gradient-based optimization of a specified loss function, progressively enhances predictive accuracy and enables the model to capture highly nonlinear relationships among variables. LightGBM is flexible, fast, and memory-efficient, making it suitable for regression, classification, and forecasting tasks across various domains.

For predicting the axial resistance of UHPC reinforced RC columns, LightGBM is particularly appropriate because the relationship between material properties, geometric parameters, and reinforcement factors with the ultimate load Pu is inherently nonlinear. Through its mechanism of sequentially learning from errors and refining predictions across many trees, LightGBM effectively models interactions among variables. Although other advanced gradient boosting variants such as XGBoost or CatBoost may offer additional optimizations, LightGBM remains a reliable, high-performance method, often employed as a benchmark in machine learning studies within structural engineering research.

**2.2.6 Cascade Forward Neural Networks (CFNN).** Cascade Forward Neural Networks (CFNN) are a variant of multilayer perceptron architectures that extend the conventional feedforward network by introducing additional forward connections [38]. Unlike traditional backpropagation neural networks (BPNNs), where each hidden layer only receives

input from the previous layer, CFNNs allow each layer to receive inputs not only from the preceding layer but also directly from the original input layer. This cascaded connection pattern enables the network to propagate raw input information throughout the entire architecture, enhancing its ability to capture both low-level and high-level feature interactions simultaneously.

The structural design of CFNNs often leads to faster learning and improved approximation capabilities, especially when dealing with highly nonlinear and complex datasets. By facilitating richer information flow across layers, CFNNs can achieve more accurate regression and prediction results with fewer hidden units compared to standard MLPs. Beyond theoretical advantages, CFNNs have been successfully applied in engineering and scientific problems, where they demonstrated robust predictive performance in scenarios requiring precise modeling of nonlinear relationships. Recent findings by Nguyen [39] revealed that the CFNN model outperformed both LightGBM and SVR in predicting the compressive strength of geopolymer concrete, particularly when data augmentation techniques were employed.

**2.2.7 Shapley additive explanation (SHAP).** Machine learning (ML) algorithms are often referred to as "black-box" models, since their internal decision-making processes are not directly visible. For this reason, enhancing model interpretability is crucial to ensure transparency, trustworthiness, and opportunities for improvement. Among the various tools used for this purpose, feature importance techniques are widely applied to evaluate the contribution of individual input variables to the prediction outcome, thereby assisting in the interpretation of ML models.

In recent years, the Shapley Additive Explanations (SHAP) framework has become one of the most widely adopted methods for explaining model behavior. SHAP creates a link between predictive accuracy and interpretability by assigning a quantitative importance score to each feature based on cooperative game theory. This enables the transformation of an opaque ML model into a more transparent, interpretable system. In the present study, SHAP is utilized to examine the most accurate predictive model and to assess the effect of each input variable on the predicted outcomes. This technique is selected because it offers both global perspectives, revealing overall feature importance across the dataset, and local insights, illustrating how features contribute to individual predictions, making it highly effective and applicable for the top-performing machine learning algorithms.

## 3. Model implementation and evaluation

### 3.1 Performance criteria

To measure the accuracy and efficiency of the machine learning (ML) models developed in this work, four performance indicators are employed: the coefficient of determination (R²), root mean square error (RMSE), mean absolute percentage error (MAPE), mean absolute error (MAE). Among these, R², RMSE, MAPE, and MAE are the most commonly used metrics in regression-based research within structural engineering. A higher R² value, approaching unity, indicates stronger predictive performance. Conversely, lower values of RMSE, MAPE, and MAE reflect better model precision. The mathematical expressions of these performance measures are presented as follows:

$$R^2 = 1 - \frac{\sum_{i=1}^{n} (y_i - \hat{y}_i)^2}{\sum_{i=1}^{n} (y_i - \bar{y}_i)^2} \tag{1}$$

$$RMSE = \sqrt{\frac{1}{n} \sum_{i=1}^{n} (y_i - \hat{y}_i)^2} \tag{2}$$

$$MAPE = \frac{1}{n} \sum_{i=1}^{n} \frac{|y_i - \hat{y}_i|}{y_i} \tag{3}$$

$$MAE = \frac{1}{n} \sum_{i=1}^{n} |y_i - \hat{y}_i| \tag{4}$$

The coefficient of determination (R²) measures how well the predicted values explain the variance in the observed data, reflecting overall goodness-of-fit. Mean absolute error (MAE) quantifies the average magnitude of prediction errors without considering their direction, providing a straightforward measure of accuracy. Mean absolute percentage error (MAPE) expresses errors as a percentage of actual values, allowing for comparison across different scales. Root mean square error (RMSE) emphasizes larger errors by squaring the deviations before averaging, offering sensitivity to outliers. By collectively analyzing these metrics, researchers can comprehensively assess predictive precision, consistency, and robustness, enabling the selection of the most reliable algorithm for a given problem.

### 3.2 Model training and test procedure

Fig 5 presents the workflow for constructing machine learning models aimed at predicting the load-bearing capacity of reinforced concrete (RC) columns strengthened with UHPC. The process involves several critical stages: assembling the experimental dataset, performing data preprocessing and refinement, training the models, optimizing hyperparameters, validating performance, and interpreting the results. Each step plays a vital role in ensuring the predictive system is both accurate and dependable.

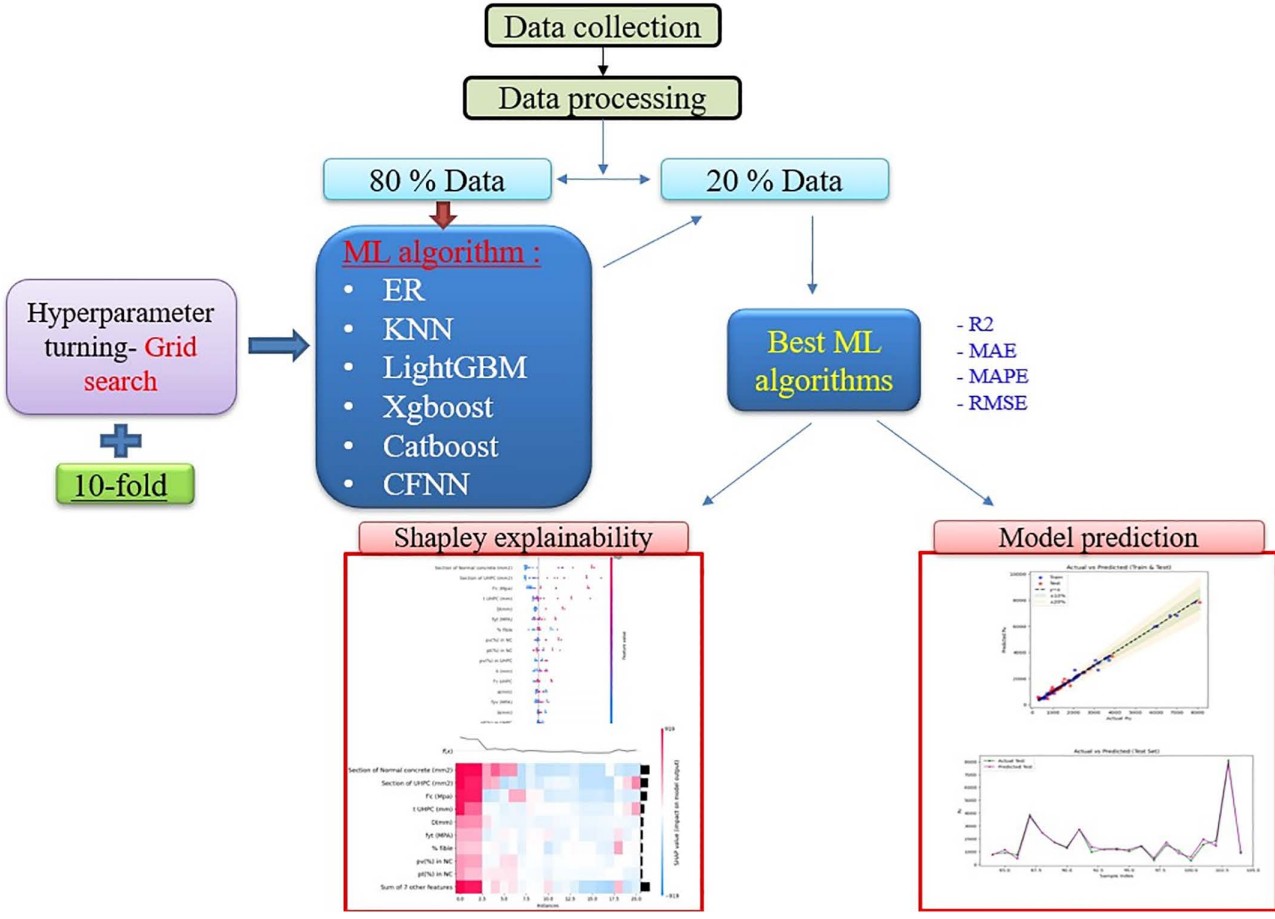

**Fig 5. Model training and test procedure.**

The dataset, compiled from prior experimental investigations on the axial capacity of UHPC-strengthened RC columns, forms the basis for model training. The data is split into training and testing subsets, with 80% used for model development and 20% held out for testing and evaluation. During training, algorithms such as ER, KNN, LightGBM, XGBoost, CatBoost, and CFNN are implemented in Python and optimized using Grid Search in conjunction with 10-fold cross-validation, allowing systematic tuning of hyperparameters while maintaining robust generalization.

Performance is measured with multiple statistical metrics, including $R^2$, RMSE, MAE, and MAPE, to comprehensively assess predictive accuracy and reliability. The algorithm demonstrating the best performance is then analyzed using SHAP values to determine the contribution of each input variable to the predicted axial load. Finally, predicted outcomes are compared directly with experimental results, providing both interpretability and validation of the machine learning framework.

### 3.3 Evaluation of model

The optimal hyperparameters obtained for each model are summarized in Table 2, which presents the tuned configurations that yielded the most favorable performance during the Grid Search process. These parameter settings reflect the specific characteristics of each algorithm and highlight the importance of hyperparameter optimization in improving predictive accuracy. Following the optimization stage, the predictive performance of all models is evaluated and benchmarked using four statistical indicators: the coefficient of determination ($R^2$), mean absolute error (MAE), mean absolute percentage error (MAPE), and root mean square error (RMSE). The comparative results, reported in Table 3, provide a comprehensive overview of how each algorithm performs in terms of accuracy, consistency, and generalization capability. This dual presentation of hyperparameter tuning (Table 2) and performance outcomes (Table 3) ensures transparency and allows a fair comparison across all models.

**Table 2. Optimal hyperparameters.**

| No | Model | Optimal hyperparameter | |
|----|-------|------------------------|---|
| 1 | ER | Min_samples_leaf | 1 |
| | | Min_samples_split | 2 |
| | | n_estimators | 100 |
| 2 | KNN | n_neighbors':3, | 3 |
| | | p | 2 |
| | | Weights | distance |
| 3 | LightGBM | Learning rate | 0.1 |
| | | colsample_bytree | 0.8 |
| | | n_estimators | 500 |
| 4 | XGboost | colsample_bytree | 1 |
| | | Learning rate | 0.01 |
| | | max_depth | 7 |
| | | n_estimators | 500 |
| | | subsample | 0.6 |
| 5 | Catboost | Depth | 4 |
| | | Iteration | 1000 |
| | | l2_leaf_reg | 5 |
| | | Learning rate | 0.05 |
| 6 | CFNN | epochs | 1000 |
| | | hidden_dim | 32 |
| | | lr | 0.001 |

**Table 3. Average predictive performance of the model obtained through K-fold cross-validation.**

| No | Model | Data Train | | | | Data Test | | | | All data | | | |
|----|-------|------|------|------|------|------|------|------|------|------|------|------|------|
| | | R² | MAE | MAPE | RMSE | R² | MAE | MAPE | RMSE | R² | MAE | MAPE | RMSE |
| 1 | ER | 0.994 | 50.31 | 3.69 | 113.05 | 0.977 | 214.77 | 22.62 | 248.42 | 0.99 | 83.20 | 7.47 | 150.22 |
| 2 | KNN | 0.994 | 43.30 | 3.11 | 112.10 | 0.974 | 228.60 | 24.86 | 262.46 | 0.99 | 80.36 | 7.46 | 154.37 |
| 3 | LightGBM | 0.974 | 169.00 | 12.39 | 241.27 | 0.945 | 253.62 | 18.87 | 384.29 | 0.97 | 185.92 | 13.68 | 275.87 |
| 4 | XGboost | 0.988 | 93.14 | 6.88 | 160.83 | 0.929 | 275.82 | 25.24 | 434.89 | 0.97 | 129.67 | 10.56 | 241.91 |
| 5 | Catboost | 0.994 | 66.32 | 5.20 | 118.44 | 0.983 | 177.18 | 19.48 | 210.66 | 0.99 | 88.49 | 8.06 | 141.76 |
| 6 | CFNN | 0.994 | 56.49 | 4.11 | 120.55 | 0.972 | 216.79 | 22.83 | 275.66 | 0.99 | 88.55 | 7.85 | 163.78 |

The comparative evaluation of the employed machine learning models provides important insights into their ability to predict the axial capacity of UHPC-jacketed RC columns. The analysis was performed using four widely accepted error indicators, namely the coefficient of determination ($R^2$), mean absolute error (MAE), mean absolute percentage error (MAPE), and root mean square error (RMSE). These indices were calculated separately for the training dataset, the unseen test dataset, and the overall dataset in order to provide a more balanced and transparent performance assessment (Fig 6).

On the training set, all six models exhibited outstanding fitting ability, with coefficients of determination ($R^2$) between 0.974 and 0.994, confirming that the selected predictors captured the majority of the variance in axial load. ER, KNN, CatBoost, and CFNN all achieved $R^2 = 0.994$, with very small errors (ER model: MAE = 50.31, RMSE = 113.05; KNN model: MAE = 43.30, RMSE = 112.10; CatBoost model: MAE = 66.32, RMSE = 118.44; CFNN model: MAE = 56.49, RMSE = 120.55). LightGBM and XGBoost also performed strongly ($R^2 = 0.974$ and 0.988, respectively), though with comparatively larger errors.

On the independent test set, model discrepancies became more evident. CatBoost model provided the best generalization, with $R^2 = 0.983$, MAE = 177.18, and RMSE = 210.66, outperforming all other approaches. ER ($R^2 = 0.977$) and KNN ($R^2 = 0.974$) also yielded competitive accuracy but with higher error levels (RMSE ≈ 248–262). LightGBM and XGBoost suffered more substantial error propagation ($R^2 = 0.929$–0.945; RMSE = 384–435), highlighting their sensitivity to overfitting despite good training performance. CFNN ($R^2 = 0.972$, RMSE = 275.66) achieved acceptable results but remained less stable than CatBoost.

When evaluated across the entire dataset, CatBoost again demonstrated the most balanced outcome ($R^2 = 0.99$, RMSE = 141.76), followed by ER ($R^2 = 0.99$, RMSE = 150.22) and KNN ($R^2 = 0.99$, RMSE = 154.37). LightGBM and XGBoost achieved slightly lower accuracy ($R^2 = 0.97$, RMSE = 241–276), whereas CFNN produced higher errors (RMSE = 163.78), confirming its relative inferiority.

In summary, CatBoost stands out as the most robust and accurate predictor for the axial capacity of UHPC-jacketed RC columns, combining strong generalization and low error. ER and KNN also delivered reliable results, while gradient boosting models showed potential but required further refinement to reduce error levels. CFNN, although effective in capturing nonlinear patterns, was consistently less competitive than ensemble-based methods

### 3.4 ML models prediction performance of Catboost

Fig 7 illustrates the comparison between the ultimate load-carrying capacity ($P_u$) predicted by the CatBoost model and the experimentally measured values for train data and test data. In the plot, the x-axis corresponds to the actual $P_u$, while the y-axis shows the predicted values. Ideally, perfect predictions would lie exactly along the diagonal line $y = x$. To visualize deviations from perfect prediction, shaded bands representing ±10% and ±20% of the actual values are included. It can be seen that most predicted points cluster close to the diagonal, with the majority falling within the ±10% range, indicating strong agreement between predicted and observed results.

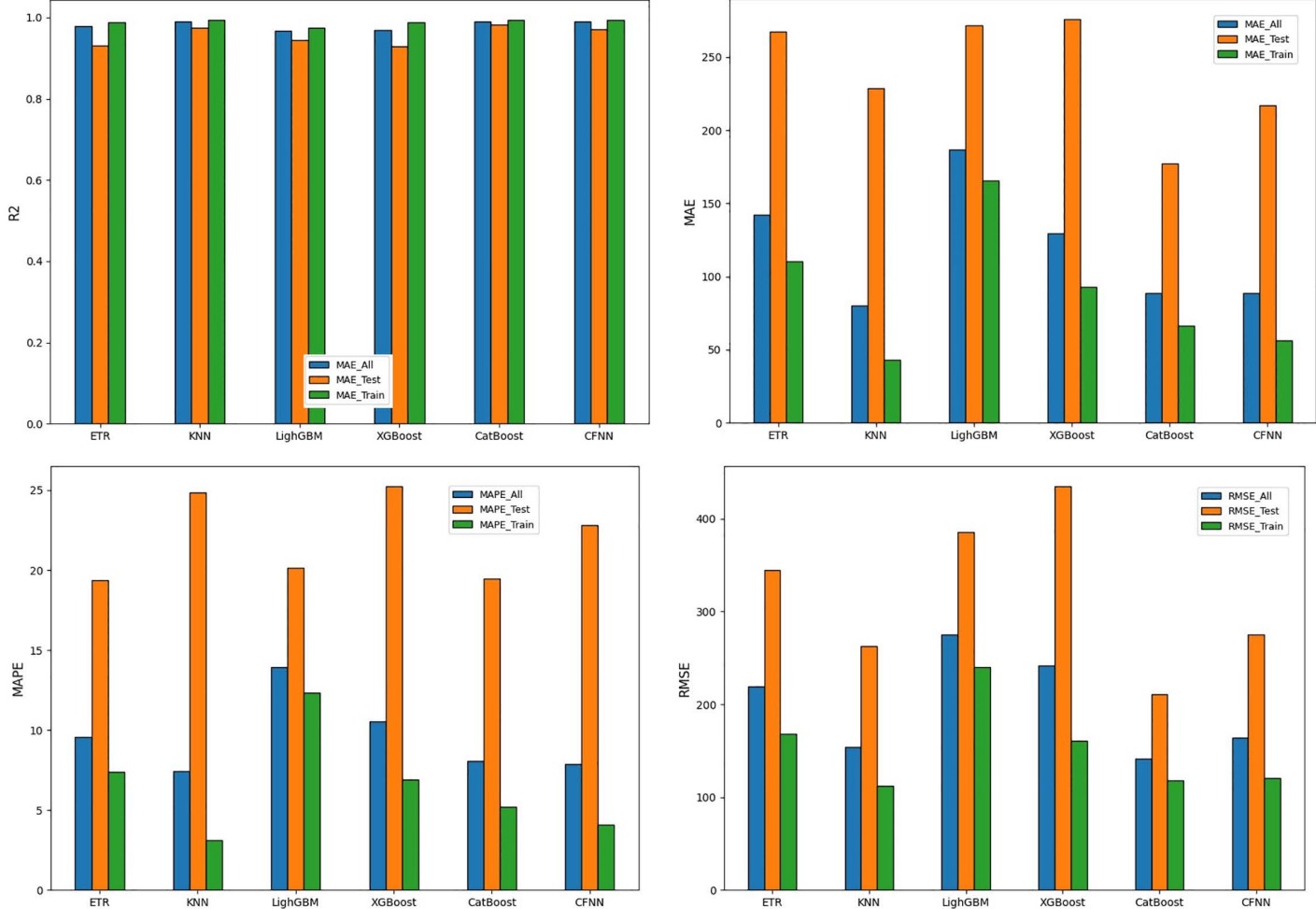

**Fig 6. Performance comparison of six models in terms of R², MAE, MAPE, and RMSE.**

As shown in the Fig 8, the horizontal axis represents the sample index, while the vertical axis shows the actual and predicted Pu values for test data. The plot exhibits a very close overlap between the predicted and actual values, indicating that the model accurately captures the underlying trend of the data.

The key performance metrics: $R^2 = 0.983$, MAE = 177.18, MAPE = 19.48 and RMSE = 210.66, which confirm the high accuracy of the predictions. Most of the predicted points lie very close to the actual points, and deviations from the actual values are minimal across the tested samples. Only a few points show larger deviations, which occur at peaks of the Pu values, but these are rare and do not significantly affect overall model performance.

Although the CatBoost model achieved a very high coefficient of determination ($R^2 = 0.983$), additional validation analyses were performed to verify that this performance did not result from overfitting. A 10-fold cross-validation confirmed the model's stability, with minimal variation in performance metrics across folds. Permutation feature importance and SHAP-based sensitivity analyses were carried out to examine the robustness and physical plausibility of the model predictions. The CatBoost model maintained stable accuracy under random feature perturbations, and the parameter influence trends agreed well with mechanical expectations, confirming that the model captures genuine structural relationships rather than memorizing the data.

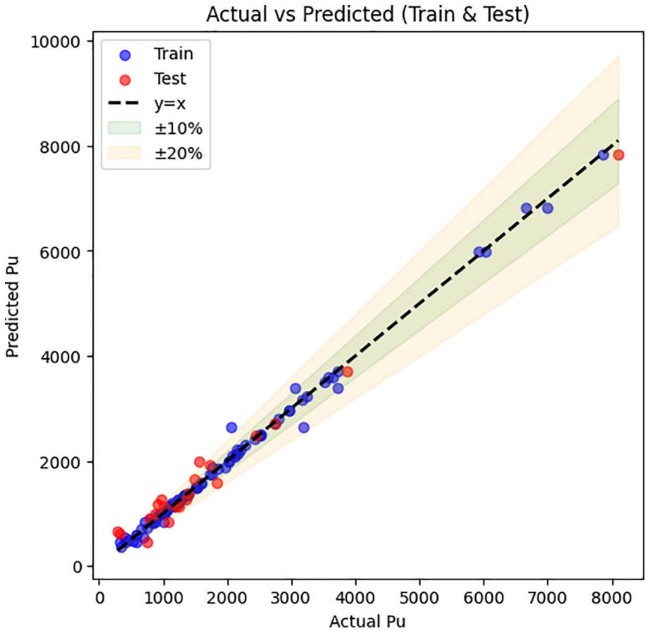

**Fig 7. Relationship between actual and predicted values with the Catboost model.**

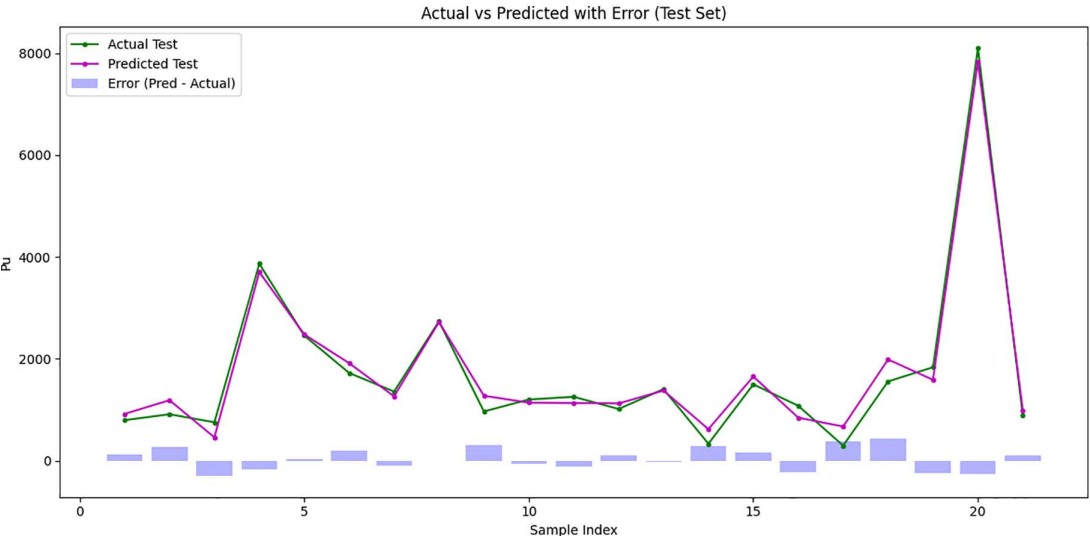

**Fig 8. Comparison of actual and predicted $P_u$ for the 20% data testing using the Catboost Model.**

Fig 8 clearly illustrates that the CatBoost model provides highly reliable predictions with low error, making it the most effective algorithm among the tested models for predicting the compressive load of UHPC columns. The high R² value combined with low MAE, MAPE, and RMSE supports the conclusion of strong predictive capability and robust generalization to unseen samples.

## 3.5 Comparison with existing calculation methods

At present, no official design code exists for evaluating the load-bearing capacity of reinforced concrete columns strengthened with ultra-high-performance concrete (UHPC). In this section, several international standards are reviewed, and their calculated results are compared with those obtained in Section 3.4. According to the general perspective of these codes, the ultimate axial load capacity, $P_u$, of a UHPC-strengthened composite column can be determined by summing the individual contributions of each constituent within the cross-section. This relationship can be expressed in the following form:

$$P_u = P_{NC} + P_{UHPC} + P_S$$

$P_{NC}$, $P_{UHPC}$ and $P_s$ denote the ultimate load capacities contributed by the core concrete, the UHPC jacket, and the longitudinal steel reinforcement, respectively. While the fundamental assumptions of the calculation approaches are generally consistent, significant variations exist in how each standard treats the contributions of the different components within a composite column. The following section reviews the procedures specified in ACI 318 [40] and Eurocode 2 [41], and evaluates their applicability to UHPC-encased reinforced concretes columns.

**3.5.1 ACI318 approach.** Within the framework of the plastic stress distribution method, the axial resistance of a UHPC-encased composite concrete column with a square cross-section is evaluated under certain simplifying assumptions regarding material behavior. Specifically, it is postulated that the longitudinal reinforcing bars have already reached their yield strength in compression at the ultimate limit state. In contrast, both the UHPC encasement and the normal-strength concrete (NSC) core are considered not to fully mobilize their compressive capacities. Instead, consistent with the reduction factors adopted in modern design standards, only 85% of their characteristic compressive strengths are taken into account. This reduction factor reflects the influence of material variability, stress non-uniformity across the section, and long-term effects such as creep and microcracking, which may prevent the section from achieving its theoretical maximum strength in practice.

On this basis, the ACI 318 standard [40] provides a design-oriented expression to quantify the axial load-bearing capacity of such columns. The formulation integrates the contributions of each component—the steel reinforcement, the UHPC jacket, and the inner concrete core—into a unified model. By summing the reduced strengths of the composite materials along with the full contribution of the yielded reinforcement, the resulting equation provides a rational and conservative estimate of the ultimate axial load capacity of UHPC-strengthened reinforced concrete columns:

$$P_u = 0.85 \left[ f'_c \cdot S_{NC} + f'_{cUHPC} \, S_{UHPC} \right] + f_y A_s$$

Where $f'_c$, $f'_{cUHPC}$ and $f'_{yt}$ are the compressive strength of the core concrete and UHPC and yielding strength of longitudinal bars; and $S_{NC}$, $S_{UHPC}$ and $A_s$ are the cross-sectional areas of UHPC encasement, core concrete, and longitudinal rebars.

**3.5.2 EC2 approach.** The Eurocode 2 (EC2) [41] provides a systematic framework for assessing the ultimate axial capacity of reinforced concrete columns, including those strengthened with UHPC jackets. Unlike ACI, EC2 explicitly incorporates the role of partial safety factors applied to both steel reinforcement and concrete, thereby reducing the nominal material strengths to design strengths. This approach accounts for uncertainties associated with material properties, construction quality, and structural analysis, ensuring a consistent level of safety and reliability across European practice.

For UHPC-encased columns with square cross-sections, EC2 assumes that the longitudinal reinforcing bars reach their design yield strength under compression, while the UHPC jacket and normal-strength concrete core are considered at their respective design compressive strengths, each modified by the appropriate partial safety factor ($\gamma$). This methodology generally produces design capacities that may differ from those obtained using ACI, depending on the values specified in the National Annex. By integrating the beneficial effects of UHPC confinement within its reliability-based design

framework, Eurocode 2 provides a balanced and rational basis for evaluating the axial load-bearing capacity of strengthened reinforced concrete columns. Accordingly, the design axial resistance in this study is determined following the EC2 formulation (Fig 9):

$$P_u = f'_c \cdot S_{NC} + f'_{cUHPC} \, S_{UHPC} + f_y A_s$$

The predicted load-bearing capacity of reinforced concrete columns strengthened with ultra-high-performance concrete (UHPC) varies considerably depending on the design standard. As shown in Table 4, the EC2 method yields an $R^2$ of 0.635, MAE of 648.972, MAPE of 37.55%, and RMSE of 924.358, indicating limited predictive accuracy. ACI 318 improve the predictions ($R^2 = 0.849$), but still exhibit considerable errors.

In comparison with the results in Table 3, this clearly illustrates that advanced machine learning algorithms can capture the complex, nonlinear behavior of UHPC-strengthened reinforced concrete columns more effectively than conventional code-based approaches. The superior accuracy of CatBoost arises from its ordered boosting, capability to handle categorical variables, and robust regularization, making it a highly reliable tool for predicting column load-bearing capacity.

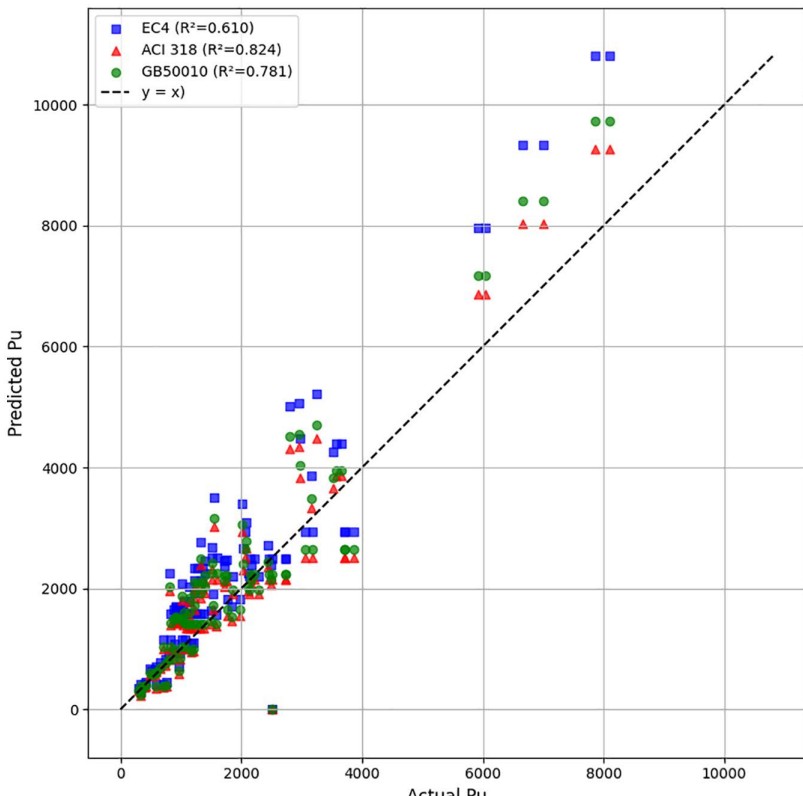

**Fig 9. Performance of EC2 and ACI 318 to predict P$_u$.**

**Table 4. Performance of model with existing calculation methods.**

| No | Method | $R^2$ | MAE | MAPE | RMSE |
|---|---|---|---|---|---|
| 1 | EC2 | 0.635 | 648.972 | 37.55 | 924.358 |
| 2 | ACI 318 | 0.849 | 446.79 | 26.623 | 594.878 |

The ML-predicted axial capacities were compared with those obtained from EC2 and ACI 318 design provisions. As expected, the design codes produced more conservative estimates due to the inclusion of global safety factors and simplified confinement models. In contrast, the ML models directly predicted the experimentally measured ultimate strengths, which correspond to mean structural capacities.

As a result, even though the ML predictions seem higher than the code-based values, this discrepancy actually indicates the lack of embedded safety margins rather than unsafe performance. For real-world applications, suitable safety or reduction factors (e.g., $\varphi = 0.75$–$0.85$) can be added to match design-level values with ML-predicted strengths. Such calibration could enable future integration of data-driven approaches into code-based design frameworks. The results thus highlight the potential of ML not as a replacement for current codes, but as a complementary predictive tool for refining design provisions and improving reliability assessment of UHPC-strengthened RC columns.

## 4. Model explain ability

### 4.1 SHAP-based analysis

The feature importance analysis (Fig 10) highlights the parameters most strongly influencing the ultimate axial load capacity ($P_u$) of UHPC- UHPC-strengthened RC columns. The sectional areas of the UHPC jacket ($S_{UHPC}$) and sectional area of normal concrete ($S_{NC}$) are identified as the most dominant features, contributing 17.24% and 15.63% of the total importance, respectively. This finding is consistent with column confinement theory, as these two parameters directly determine the load-bearing capacity of the composite cross-section. A larger UHPC jacket area and concrete core area provide enhanced confinement and greater axial stiffness, leading to a substantial increase in the ultimate axial load capacity.

The compressive strength of normal concrete ($f_c'$) ranks third (11.25%), followed by UHPC jacket thickness ($t_{UHPC}$) with 8.58%. Both features play vital roles in improving the overall confinement and stiffness of the strengthened column. The diameter (D) and column height (h) also exhibit notable contributions (6.96% and 5.76%), reflecting their influence on the slenderness ratio and global stability under axial compression.

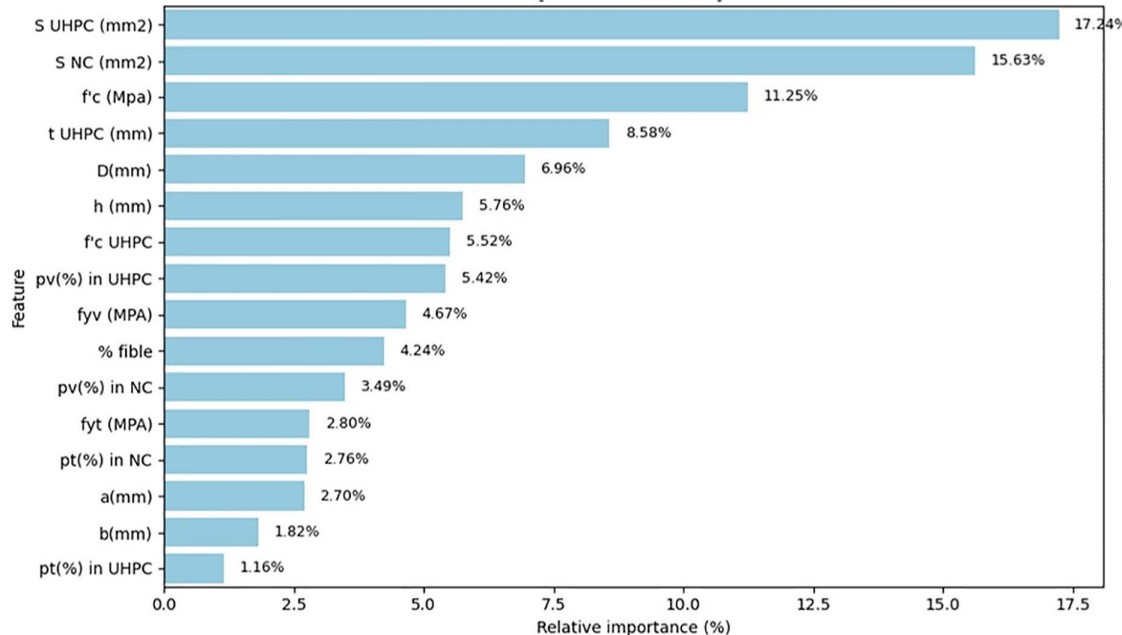

**Fig 10. Relative importance of input features for UHPC-confined RC columns.**

The UHPC compressive strength ($f'_{c\,UHPC}$) and transverse reinforcement ratio in the UHPC layer ($p_v$ in UHPC) contribute moderately (5.52% and 5.42%), underscoring the importance of material strength and lateral confinement in enhancing ductility and preventing premature failure. Yield strength of transverse reinforcement ($f_{yv}$) and fiber content (% fiber) also show measurable effects, indicating that higher reinforcement strength and appropriate fiber dosage improve the ultimate axial load capacity.

Parameters such as the longitudinal reinforcement ratio ($p_t$) and transverse reinforcement ratio ($p_v$) in normal concrete, as well as the column length (a), columns width (b), and longitudinal reinforcement ratio in UHPC ($p_f$ in UHPC), show lower relative importance (below 4%), suggesting their effects are secondary or act synergistically with other key parameters.

Overall, the analysis demonstrates that the axial capacity of UHPC-confined RC columns is primarily governed by the sectional areas and compressive strengths of both the core and the UHPC jacket, followed by geometric parameters and reinforcement detailing. These insights confirm the strong alignment between data-driven findings and structural confinement mechanisms, reinforcing the physical interpretability.

## 4.2 Feature dependency analysis

The SHAP feature dependency analysis provides further insights into how variations in individual parameters affect the predicted axial capacity of UHPC-jacketed RC columns. Fig 11 illustrates the SHAP values for each feature, with the color scale indicating relative feature magnitudes (red = higher values, blue = lower values). Consistent with the feature

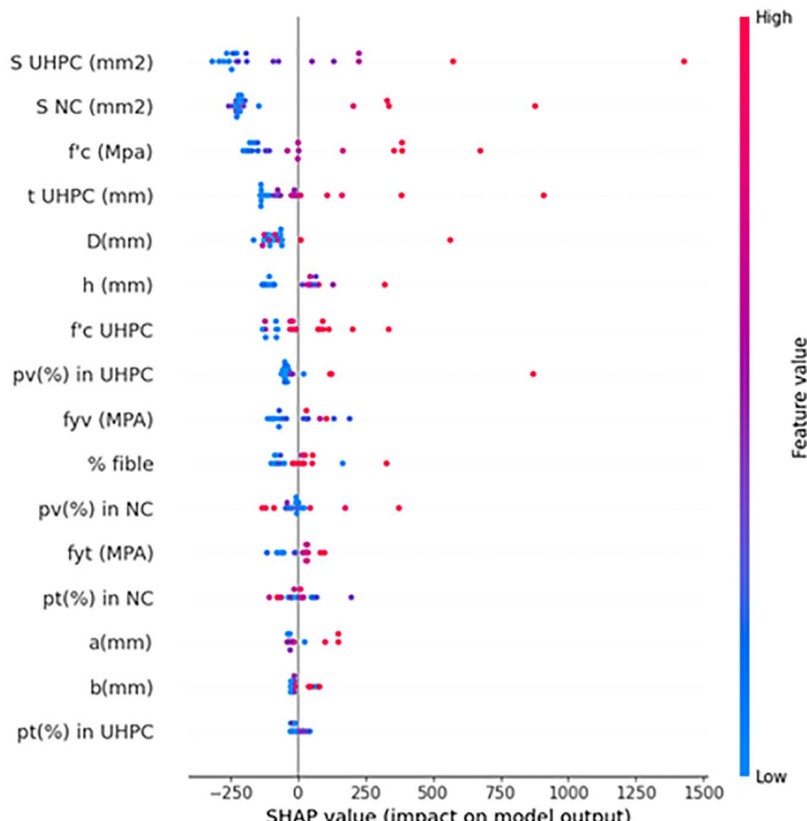

**Fig 11. SHAP dependency plot for UHPC-confined RC columns.**

importance ranking, the cross-sectional areas of UHPC and normal concrete ($S_{UHPC}$, $S_{NC}$) exert the most significant influence. Larger section sizes (red points) are associated with strongly positive SHAP values, confirming their direct contribution to axial strength by increasing the effective load-bearing area.

Material properties also show clear trends. Higher compressive strength of normal concrete (f'c) and greater UHPC thickness ($t_{UHPC}$) correspond to positive SHAP contributions, emphasizing the dual role of core quality and jacket confinement in strength enhancement. Similarly, increases in UHPC compressive strength ($f'_{c\,UHPC}$) and reinforcement ratios ($p_v$% in UHPC and NC) generally improve predictions, though some scatter suggests interaction effects with other geometric parameters.

Reinforcement-related factors, such as yield strength of longitudinal bars ($f_{yt}$) and transverse reinforcement ($f_{yv}$), exhibit moderate but consistent positive impacts, indicating that higher steel strength enhances confinement and load resistance. In contrast, fiber volume fraction (% fiber) shows a more dispersed pattern, with both positive and negative SHAP values, implying that its effect depends on dosage—moderate fiber contents contribute positively by improving crack control, while excessive amounts may reduce workability and efficiency.

Geometric factors such as column diameter (D), height (h), and column length (a) also play secondary but relevant roles, with SHAP values reflecting their influence on load transfer mechanisms. Meanwhile, parameters like $p_{t\,(\%)}$ in UHPC or column width (b) demonstrate relatively limited influence, aligning with their lower overall importance.

The SHAP analysis revealed that the UHPC jacket compressive strength and jacket thickness have the strongest positive influence on the predicted axial load capacity. This is consistent with classical confinement theory, where a thicker and stronger jacket provides higher lateral pressure, enhancing the confined core concrete strength. The positive contributions of the core concrete strength and longitudinal reinforcement ratio also align with analytical confinement models, confirming that the ML framework correctly learns the combined effects of material strength and reinforcement on load resistance. Conversely, geometric parameters such as slenderness ratio exhibit negative SHAP values, reflecting the reduction in stability and axial strength observed in structural mechanics. These physically interpretable trends confirm that the proposed ML models, particularly CatBoost, not only provide accurate predictions but also capture the governing mechanical behaviors of UHPC-confined RC columns.

Overall, the SHAP dependency plots confirm that structural capacity is most strongly governed by cross-sectional dimensions and material strength, while reinforcement and fiber characteristics provide additional but more variable contributions. These findings are consistent with engineering mechanics, where geometry and material quality dominate axial resistance, and detailing factors modulate the overall response.

## 4.3 ICE and PDP

The ICE (Individual Conditional Expectation) and PDP (Partial Dependence Plot) analyses (Fig 12) further validate the interpretability of the CatBoost model by illustrating the nonlinear and, in several cases, quasi-monotonic relationships between the key predictors and the predicted axial load capacity ($P_u$). Specifically, increases in the cross-sectional reinforcement areas ($S_{NC}$ and $S_{UHPC}$) consistently lead to higher predicted capacities, confirming their dominant structural contribution. Similarly, enhancements in UHPC compressive strength ($f'_{c\,UHPC}$) produce incremental gains, though the overall sensitivity remains moderate. In contrast, parameters such as fiber volume fraction (% fiber) and UHPC cover thickness ($t_{UHPC}$) exhibit relatively minor influence, indicating that their effects are secondary and may depend on interaction with other design variables. Overall, these findings suggest that the reinforcement configuration and column geometry serve as the governing determinants of load capacity, while UHPC primarily functions as a supplementary strengthening layer that improves stiffness and stress distribution efficiency rather than directly governing axial resistance.

## 5. Limitation and future recommendations

One important limitation of this study concerns the scope and diversity of the dataset employed. The current database does not fully capture the wide variation in column geometries, reinforcement layouts, or concrete strength grades. As

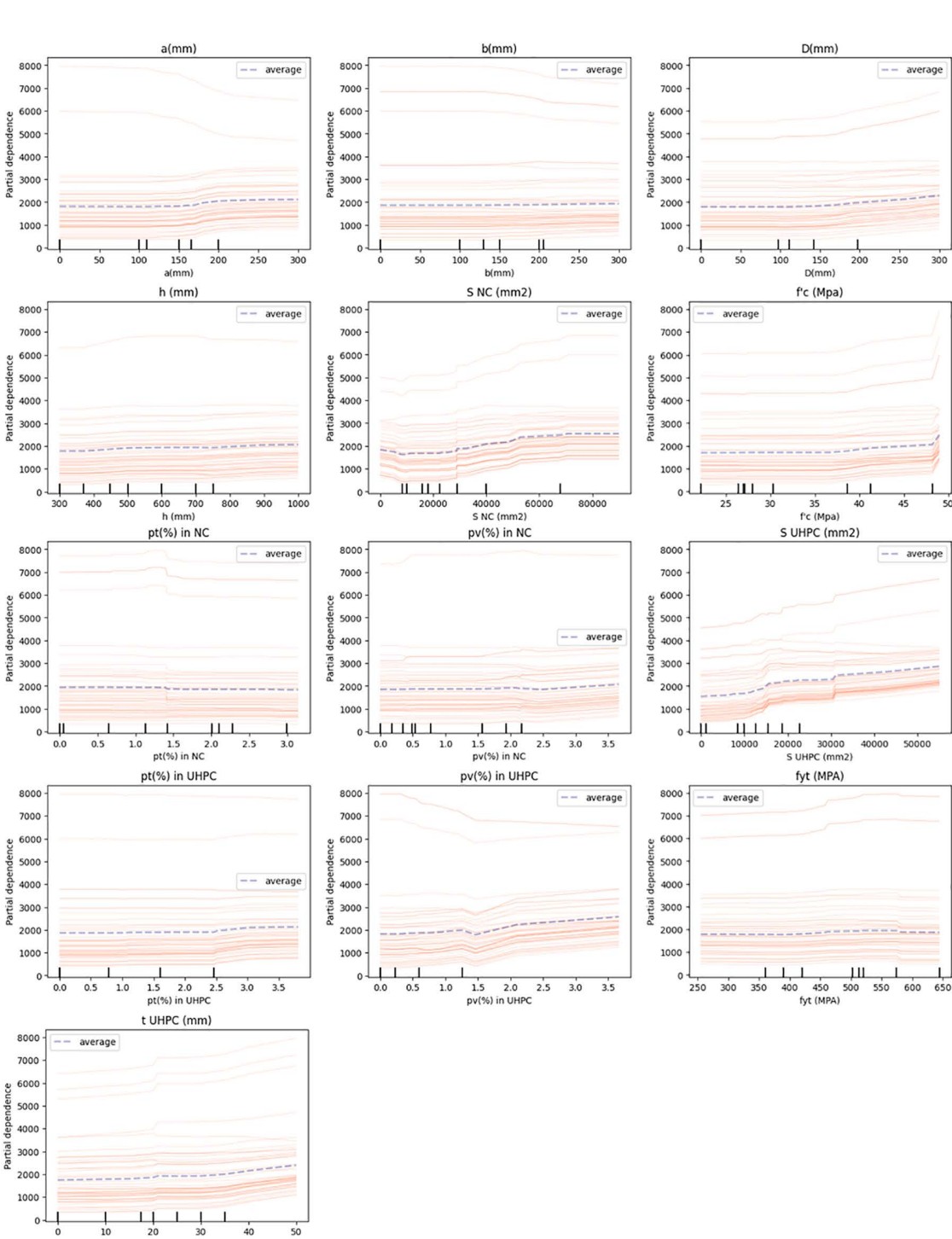

**Fig 12. ICE (Individual Conditional Expectation) and PDP – (Partial Dependence Plot) analyses.**

outlined in Section 2, because only a small number of column tests are currently available, the investigation was restricted to UHPC–RC composite columns, which narrows the applicability of the model to this structural type. In addition, the distribution of input parameters, shown in Fig 2, is uneven, leaving some ranges underrepresented. This imbalance can reduce predictive reliability when the model is applied to column cases that deviate from the observed dataset.

Overcoming these issues will require extending the dataset to cover a broader set of structural scenarios for UHPC–RC columns. Although incorporating results from previous experimental work, as done in this research, helps to enrich the data, it is both labor-intensive and vulnerable to inconsistencies across testing protocols and reporting practices. A promising solution would be to establish a shared, open-access repository dedicated to UHPC–RC composite column research, enabling standardized data exchange and collaboration across the community. Such a resource would support faster data accumulation and improve the generalizability of predictive models. Additionally, data augmentation methods—such as synthetic data generation, advanced interpolation, or generative modeling—offer a means of filling gaps in underrepresented ranges. Future studies could also integrate metaheuristic optimization techniques—such as Particle Swarm Optimization (PSO) [42], Grey Wolf Optimizer [43], …—to fine-tune model hyperparameters, enhance predictive accuracy, and better capture complex nonlinear behaviors in UHPC–RC columns.

The machine learning models developed in this study were trained exclusively on experimental data of RC columns strengthened with UHPC jackets. As such, their predictive validity is limited to similar structural configurations. Although the input parameters span a wide range of geometric and material properties, applying the model to other strengthening techniques would require retraining with relevant datasets. However, the proposed ML framework is flexible and can be easily extended to such cases once adequate data becomes available. The same feature engineering and interpretability procedure can be utilized to develop specialized predictive tools for various strengthening systems and geometries.

## 6. Conclusions

This study comprehensively evaluated the predictive performance of several advanced machine learning (ML) models for estimating the ultimate axial load capacity ($P_u$) of reinforced concrete (RC) columns strengthened with ultra-high-performance concrete (UHPC) jackets. Six ML models — Extremely Randomized Trees (ER), k-Nearest Neighbors (KNN), Light Gradient Boosting Machine (LightGBM), eXtreme Gradient Boosting (XGBoost), CatBoost, and Cascade Forward Neural Network (CFNN) — were developed and validated using an experimental database comprising 105 test results. The findings highlight the capability of data-driven methods to accurately capture the complex nonlinear interactions among geometric, material, and reinforcement parameters governing the axial behavior of UHPC–strengthened columns.

Among the evaluated models, CatBoost demonstrated the best overall performance, achieving an $R^2$ of 0.983 on the test set and an overall $R^2$ of 0.99 with a remarkably low RMSE of 141.76 kN, MAE of 88.49 kN, and MAPE of 8.06%. This reflects its strong generalization and robustness in modeling nonlinear relationships. The ER and KNN models followed closely with comparable $R^2$ values of 0.99, though their prediction errors were slightly higher (RMSE from 150 to 154 kN). On the other hand, LightGBM and XGBoost achieved satisfactory performance on the training data ($R^2 = 0.97$–0.99) but exhibited more noticeable drops in testing accuracy ($R^2 = 0.929$–0.945, RMSE = 384–435 kN), indicating mild overfitting. The CFNN model, while achieving a respectable $R^2$ of 0.972 on the test set, produced higher RMSE values (275.66 kN), suggesting a relatively weaker ability to generalize compared with ensemble-based methods.

The SHAP-based feature interpretation of the CatBoost model provided valuable insights into the governing parameters affecting axial capacity. The results revealed that the cross-sectional areas of normal concrete (NC) and UHPC, along with the compressive strength of NC, were the most influential predictors. Secondary factors such as reinforcement ratio, stirrup strength, and fiber content contributed primarily to ductility and confinement effects rather than peak strength. These findings align with established structural mechanics principles, reinforcing the interpretability and reliability of the ML-based framework.

When compared with traditional design, the superiority of the ML approach becomes even more pronounced. The EC2 and ACI 318 equations achieved R² values of 0.635 and 0.849, respectively, with significantly higher RMSE values (924.36 kN and 594.88 kN). In contrast, the CatBoost model reduced the RMSE to just 141.76 kN, representing a reduction in prediction error of approximately 75–85% relative to conventional design methods. This substantial improvement highlights the limitations of current code-based formulations, which rely on simplified assumptions, and underscores the potential of ML models to provide more accurate and generalizable predictions for UHPC–strengthened RC columns.

Beyond predictive performance, the proposed ML framework offers a data-driven foundation for future code calibration. The model can determine the most important factors influencing load-carrying capacity by examining parameter interactions and feature importance. This information can then be used to improve empirical coefficients or partial safety factors in upcoming updates of ACI and EC2 provisions. Additionally, by emphasizing parameter ranges or combinations that have the biggest impact on structural behavior, the interpretability results can direct focused experimental programs, increasing experimental efficiency.

This study demonstrates that ensemble learning models, particularly CatBoost, offer a reliable and interpretable framework for predicting the axial capacity of UHPC-confined RC columns under limited experimental data conditions. The integration of SHAP analysis further enhances the transparency of the predictions, enabling deeper physical understanding of parameter influence. By combining data-driven intelligence with fundamental mechanics, this approach provides both a scientific foundation and practical guidance for designing and optimizing UHPC strengthening systems. Future work should focus on expanding the experimental database, incorporating additional loading scenarios, and exploring hybrid ML–mechanics-based models to further enhance the robustness and applicability of the proposed framework.

## Supporting information

**S1 Data. SI data 1.**
(CSV)

## Author contributions

**Conceptualization:** Viet Hai Hoang.

**Data curation:** Viet Hai Hoang, Van Thuc Ngo.

**Formal analysis:** Viet Hai Hoang, Van Thuc Ngo.

**Funding acquisition:** Viet Hai Hoang.

**Investigation:** Viet Hai Hoang, Minh Quang Tran, Van Thuc Ngo.

**Methodology:** Viet Hai Hoang, Minh Quang Tran.

**Supervision:** Viet Hai Hoang, Van Thuc Ngo.

**Validation:** Viet Hai Hoang, Van Thuc Ngo.

**Visualization:** Viet Hai Hoang, Minh Quang Tran.

**Writing – original draft:** Viet Hai Hoang, Minh Quang Tran, Van Thuc Ngo.

**Writing – review & editing:** Viet Hai Hoang, Minh Quang Tran, Van Thuc Ngo.

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
