## [Decision Letter · Decision Letter 0]

27 Oct 2025

Dear Dr. Hoang,

Thank you for submitting your manuscript to PLOS ONE. After careful consideration, we feel that it has merit but does not fully meet PLOS ONE’s publication criteria as it currently stands. Therefore, we invite you to submit a revised version of the manuscript that addresses the points raised during the review process.

We look forward to receiving your revised manuscript.

Kind regards,

Parthiban Kathirvel

Academic Editor

PLOS ONE

Journal Requirements:

“This research is funded by the University of Transport and Communications (UTC) under grant number T2025-CT-003TD”

4. Thank you for stating the following in your manuscript:

“This research is funded by the University of Transport and Communications (UTC) under grant number T2025-CT-003TD”

“This research is funded by the University of Transport and Communications (UTC) under grant number T2025-CT-003TD”

Reviewers' comments:

Reviewer's Responses to Questions

**Comments to the Author**

1. Is the manuscript technically sound, and do the data support the conclusions?

Reviewer #1: Yes

Reviewer #2: Partly

Reviewer #3: Yes

2. Has the statistical analysis been performed appropriately and rigorously?

Reviewer #1: Yes

Reviewer #2: Yes

Reviewer #3: Yes

3. Have the authors made all data underlying the findings in their manuscript fully available?

Reviewer #1: Yes

Reviewer #2: Yes

Reviewer #3: Yes

4. Is the manuscript presented in an intelligible fashion and written in standard English?

Reviewer #1: Yes

Reviewer #2: Yes

Reviewer #3: Yes

Reviewer #1: 1. Can you please check the model number in line 118 – 119, are they six or seven Models or seven the “ for predictive modeling, seven advanced algorithms widely adopted in structural engineering were utilized, namely ER, KNN, LightGBM, XGBoost, CatBoost, and CFNN “

2. Can you please check Table 2, Under column “No” are the model number 6 or 7. It might be typing mistake. Where is model number 3.

3. If possible, please consider Adding Error Bars in Figures 3 & 6.

4. After Figure 7, It was writing Figure 3 above “ Actual vs Predicted with Error (Test Set) Diagram “ , Can you please check, I think this should be Figure 8.

5. Discuses Why Authors only selected 6 models in this study. Are there other models that you may be included.

6. There are already many articles of this kind; what is the innovation of this work, and it should be highlighted in the abstract.

7. Since there are numerous machine learning algorithms, the introduction of algorithms in this paper should be appropriately simplified

8. The machine learning algorithms used in this study cannot produce explicit expressions, which makes practical application inconvenient. How should this issue be addressed?

9. How was the database constructed? What were the principles for data exclusion

10. Include number of data in abstract

11. Rewrite the research gap

12. Comparison of previous findings with current study is missing in conclusion part.

13. To enhance the quality of manuscript author may consider following references

a https://doi.org/10.3390/ma15124164

b. https://doi.org/10.1016/j.jmrt.2023.03.037

c. https://doi.org/10.1016/j.asej.2023.102548

d. https://doi.org/10.3390/ma15155232

e. https://doi.org/10.1371/journal.pone.0322947

Reviewer #2: The paper has good novelty and relevance, particularly in combining explainable machine learning with UHPC-strengthened RC columns; however, it still requires revision before being considered for publication.

1. The database compilation process needs more technical transparency. The authors mention collecting 105 tests from fourteen studies, but it is unclear how data inconsistencies, unit conversions, and missing parameters were handled.

2. The justification for selecting the six machine learning models appears superficial. The rationale behind excluding other established methods (e.g., Random Forest, SVR, or hybrid deep learning models) should be better articulated.

3. The model evaluation lacks statistical robustness. The study heavily relies on R², MAE, RMSE, and MAPE without performing residual diagnostics, uncertainty quantification, or k-fold variation analysis.

4. The reported R² of 0.99 for both training and overall datasets raises concerns about potential overfitting. The authors should include a sensitivity or permutation-based validation to confirm the reliability of such high performance.

5. The paper does not discuss the physical interpretability of the ML outputs in sufficient depth. Although SHAP is used, the connection between feature importance and actual confinement mechanics could be better substantiated with structural theory.

6. The comparison with EC2 and ACI 318 is too simplistic. The authors should discuss how their ML-predicted loads relate to safety margins or design factors, and whether these predictions can be realistically implemented in design practice.

7. The dataset’s limited scope (mostly UHPC-jacketed columns) restricts the generalization of findings. The authors should clarify whether their model can be applied to other strengthening configurations or geometries.

8. Figures and tables lack sufficient reference in the text, and some captions are overly descriptive…

9. The conclusion section repeats earlier results but does not propose how this framework can guide future code development or experimental design.

10. Ttypographical and grammatical inconsistencies appear in sections 2 and 3.

11. References, while adequate, could include a few more recent studies on ML-aided structural design beyond 2025 to strengthen the literature foundation.

Reviewer #3: This research develops a database on RCC columns strengthened by ultra-high-performance concrete to establish the machine learning framework to predict the ultimate axial load capacity using six different models.

This is good research highlight the capability of advanced ML to capture complex nonlinear effects more effectively than traditional methods.

**Do you want your identity to be public for this peer review?** For information about this choice, including consent withdrawal, please see our Privacy Policy

Reviewer #1: **Yes: ** A.Alzlfawi

Reviewer #2: No

Reviewer #3: **Yes: ** Tejwant Singh Brar

---

## [Author Response · Author response to Decision Letter 1]

8 Nov 2025

We would like to thank the reviewers for their careful and thorough reading of this manuscript and for the thoughtful comments and constructive suggestions, which have helped us improve the manuscript. Our response is as follows (the amendments are highlighted in green or marked in red in the revised manuscript)

---

## [Decision Letter · Decision Letter 1]

11 Dec 2025

Machine learning-based prediction of the axial load capacity of UHPC strengthened reinforced concrete columns: A comparative analysis

PONE-D-25-53881R1

Dear Dr. Hoang,

We’re pleased to inform you that your manuscript has been judged scientifically suitable for publication and will be formally accepted for publication once it meets all outstanding technical requirements.

Kind regards,

Parthiban Kathirvel

Academic Editor

PLOS One

Reviewers' comments:

Reviewer's Responses to Questions

**Comments to the Author**

Reviewer #1: All comments have been addressed

Reviewer #3: All comments have been addressed

2. Is the manuscript technically sound, and do the data support the conclusions?

Reviewer #1: Yes

Reviewer #3: Yes

3. Has the statistical analysis been performed appropriately and rigorously?

Reviewer #1: Yes

Reviewer #3: Yes

4. Have the authors made all data underlying the findings in their manuscript fully available?

Reviewer #1: Yes

Reviewer #3: Yes

5. Is the manuscript presented in an intelligible fashion and written in standard English?

Reviewer #1: Yes

Reviewer #3: Yes

Reviewer #1: Thank you for the opportunity to review this manuscript. I highly recommend publishing this paper. The study addresses an important topic and offers valuable contributions to the existing body of knowledge. The methodology is sound, the results are clearly presented, and the conclusions are well supported by the findings. Overall, the manuscript is well organized, coherent, and demonstrates a high level of scientific rigor.

Reviewer #3: All the suggestions and corrections suggested have been incorporated in the revised manuscript so the manuscript can be accepted for publication.

**Do you want your identity to be public for this peer review?** For information about this choice, including consent withdrawal, please see our Privacy Policy

Reviewer #1: **Yes: ** Abdullah Alzlfawi

Reviewer #3: **Yes: ** Tejwant Singh Brar

---

## [Editor Report · Acceptance letter]

PONE-D-25-53881R1

PLOS One

Dear Dr. Hoang,

I'm pleased to inform you that your manuscript has been deemed suitable for publication in PLOS One. Congratulations! Your manuscript is now being handed over to our production team.

Kind regards,

on behalf of

Dr. Parthiban Kathirvel

Academic Editor

PLOS One